# "Almost Like Family. Or Were They?" Vikings, Frisian Identity, and the Nordification of the Past

## Simon Halink

Department of Historical Research, Fryske Akademy, 8911 DX Leeuwarden, The Netherlands; shalink@fryske-akademy.nl

**Abstract:** In the course of the twentieth century, the glorified image of Viking Age Scandinavia exerted an increasing attraction on intellectuals and nation builders in remote parts of Europe, especially those which self-identified as peripheral, marginalized, and 'northern'. In the Dutch province of Friesland, the cultivation of a Frisian national identity went hand in hand with an antagonizing process of self-contrastation vis-à-vis the urbanized heartland in the west of the country. Fueled by these anti-Holland sentiments, the adoption of Nordic identity models could serve to create alternative narrative molds in which to cast the Frisian past. In this article, I will chart this process of cultural "nordification" from its initial phase in the writings of Frisian Scandinavophiles to contemporary remediations of Frisian history in popular culture and public discourses. In this context, special attention will be paid to the reception history of the pagan King Redbad (d. 719) and his modern transformation from 'God's enemy' to beloved national icon.

**Keywords:** Friesland; Vikings; popular culture; national identity; regional identity; borealism; minorities; images of the north

## 1. Introduction

From October 2019 to March of the following year, the Frisian Museum ('Fries Museum') in Leeuwarden, capital of the Dutch province of Friesland (Frisian: Fryslân), hosted an exhibition on the Viking Age in *Frisia*, as the extended coastal region in the north and west of the Low Countries was known in the Early Middle Ages. The exhibition, which displayed not only archaeological finds from the region, but also prestigious artifacts on loan from both Dutch and foreign museums, received favorable reviews and large numbers of visitors.[1] Although Friesland is not generally considered part of the Scandinavian 'heartland' of Viking culture, the title of the exhibition, *Wij Vikingen* (We Vikings), was selected for the purpose of highlighting the strong cultural ties and shared characteristics linking Scandinavia to Frisia. The Frisians had not only been victims of the Viking incursions; they frequently participated in those raids themselves. According to Diana Spiekhout, curator at the Frisian Museum, the aim of the exhibition was to portray the Vikings not as an ethnic category, but as the pirates they really were; a sort of "Hells Angels, waarbij je kon aansluiten of niet [...] Die werkelijkheid was toen veel genuanceerder. Niet iedere Scandinaviër was Viking, en echt niet iedere Viking was Scandinaviër" (Spiering 2019; Janssen 2020, p. 50; Hells Angels, which you could either join or not [...] That reality was much more nuanced back then. Not every Scandinavian was a Viking, and certainly not every Viking was a Scandinavian).[2] In other words, the Vikings were never only 'them', the others; they could also be 'us' or the Frisian ancestors themselves.

However, this nuanced point of departure is somewhat lost in the sensationalist opening words of the blurb on the back of the exhibition catalogue, which, instead, caters to a popular and long-standing sense of kinship with the High North: "Vikingen en Friezen. Bewoners van de Noordzeekusten, blond en stoer en met een gemene deler op het gebied van taal en cultuur. Bijna familie toch, of niet?" (Stoter and Spiekhout 2019, blurb; Vikings

and Frisians. Inhabitants of the North Sea coasts. Blond and sturdy, with shared cultural characteristics and a similar language. Almost like family. Or were they?). Although the wording is ambiguous, this phrase ties into the Frisian tradition of ethnic and cultural identification with Scandinavia, the myth of the 'Frisian Viking', which can be traced back to the nineteenth century (Janssen 2020, p. 46; IJssennagger-van der Pluijm 2021). This myth is very much alive and frequently makes its appearance in the media when, for instance, regional politicians serving the interests of the Frisians—posing as Frisian Vikings—are at loggerheads with the central government in The Hague (IJssennagger-van der Pluijm 2021).

In the present article, I will chart the historical development leading to the 'nordification' of the Frisian self-image since the nineteenth century, and I will critically assess the contemporary manifestations of this cultural conflation in popular culture and political narratives. How did the figure of the Viking become reimagined and repurposed as the embodiment of primordial Frisianness in the twenty-first century? How has the preceding cultivation of Romantic images of the North paved the way for this development? And in what ways is the Viking stereotype (Birkett and Dale 2020) mobilized as a narrative template (Wertsch 2008) in contemporary identity discourses and nativist fantasies of former greatness? In addressing these issues, special attention will be paid to the nexus of reciprocal links between cultural memory, academia, political activism, the experience industry, and popular culture.

Finally, this contribution will reflect on the role of the scholar in all this. Is it our task to counterbalance these popular and oftentimes ideologically charged and distorted images of the past with solid scholarship? Or does our mission lie somewhere else? In the concluding reflections, I will address the "importance of being aware of the social task of scholarship", as laid out by Jan Alexander van Nahl in his introductory paper to the present special issue (van Nahl 2022, p. 1).

## 2. Shifting Concepts of Northernness

In recent years, the classical dichotomy of 'heathen Viking invaders' on the one hand, and 'Christian, continental victims' on the other, has been debunked and corrected by the careful examination of both material and written source material from the Early Middle Ages. The cultural and economic ties linking the coastal region of what would in later centuries become known as the Netherlands to Viking Age Scandinavia, were much stronger than previously imagined, and it is therefore justifiable to designate the whole of northern and northwestern Europe as a single 'North Sea culture': a shared cultural sphere or cultural continuum (IJssennagger 2013, 2017).

Frisia—not to be confused with the current, much smaller province of Friesland/Fryslân in the north of the Netherlands[3]—stretched along the coast roughly from the estuary of the Weser (now in Germany) in the north to the Zwin near Bruges (now in Belgium) in the south[4], with the region around Utrecht as its supposed center of power. Although much remains unknown about the political structure of Frisia, the medieval sources indicate that there was a lineage of heathen rulers or kings who may have ruled this area (or parts of it) from the mid-seventh century to 734, when the Frisians were defeated at the Battle of the Boarn and their lands became part of the Frankish Empire. So, by the time the first Viking ships appeared on the Frisian coast, early in the ninth century, Frisia had only very recently been incorporated by the Carolingians, and the process of Christianization was far from completed yet. In that sense, Frisia constituted an area in transition: a liminal space between continental Christendom—with its ecclesiastical power structure—and the heathen North, where the structural connections with Scandinavia had not yet been severed and where 'pagan sympathies' may very well have flared up and driven some of its inhabitants back into the arms of a pre-Christian Germanic faith and onto the Viking longships (IJssennagger 2013, 2017). It was only after the Viking Age that the Frisian lands were thoroughly and successfully Christianized and that Frisians began to regard themselves as a pious people in harmonious coexistence with the other Christian nations of Europe. In the medieval imagination, the North was generally perceived in negative terms, as the *grima herna* (Old

Frisian for 'grim corner') of Europe and associated with unfavorable characteristics such as barbarism and paganism. Or, as Alcuin famously summarized the sentiment in the eighth century: *Omne malum ab aquilone* (Popkema 2013; All evil hails from the north).

The history of the Frisian kings, and that of King Redbad (d. 719) in particular—who successfully fended off Charles Martel and his Franks at Cologne and famously refused baptism with one foot already in the baptismal font upon hearing that, as a Christian, he would never be joined with his pagan ancestors residing in Hell[5]—was now considered a shameful blemish on the reputation of Christian Frisia. In order to remedy this situation, Frisian chroniclers distanced themselves from this pagan ruler by portraying Redbad not as a Frisian but as a Danish *intruder*: a Viking-style pagan invader from the North, who enslaved his Frisian subjects and forced his tyrannical rule upon them (Halink 2020, pp. 106–7; Meeder and Goosmann 2018, pp. 160–66; Bremmer 2020, p. 26). In some traditions, the Danes are even blamed for reverting the Frisians—apparently already Christianized at an earlier stage!—to paganism (Nijdam 2009, p. 60). In this historical narrative, expressed in several *chansons de geste* from the eleventh century onwards, Charlemagne (who in reality was born almost thirty years after Redbad's death) is hailed as the liberator of Frisia, who defeated Redbad and in some instances even transformed him into a respectable member of his courtly entourage (Nijdam and Knottnerus 2019, pp. 95–98). The North/South dichotomy centered around these prominent memory figures, also expressed in the corpus of medieval Frisian law texts in the vernacular, is interpreted by Nelleke IJssennagger as follows:

> The transition of Frisia into the southern sphere of influence is attributed to the South, personified in Charlemagne, the most famous of the Carolingian kings, in the Frisian laws. The terrible, heathen North that Frisians once belonged to similarly becomes personified by the most famous of the Frisian 'kings', the heathen Redbad or Radbod. Both are quite logical. (IJssennagger 2017, p. 132)

By reimagining Redbad as a "heathen subjugator and bully from Denmark" (Bremmer 2020, p. 26), the negative half of a binary opposition, Frisia's Frankish identity and its dissociation from northern Europe were reiterated. Throughout the later Middle Ages and the Early Modern period, Frisian chroniclers would apply Biblical narrative templates to their renditions of the Frisian past, presenting their nation as a 'chosen people' modeled on the Israelites (Mol and Smithuis 2021).

The great paradigm shift in the way the North was perceived occurred in the second half of the eighteenth century, after the 'rediscovered' poems of the 'Celtic bard' Ossian (first published in 1761 by their 'discoverer' James Macpherson) triggered a Europe-wide interest in all things Celtic and Nordic; the distinction between these two categories was still quite blurry at that time. In the wake of this Celtic *vogue*, and even after the Ossian poems had been exposed as forgeries, intellectuals in northern and western Europe grew increasingly interested in the great antiquity of their own cultures, which were by now no longer considered inferior to the much-revered Classical cultures of southern Europe (Leerssen 2005, 2019). The emancipation of northern culture was further encouraged by prominent German intellectuals such as Johann Gottfried von Herder—who urged German artists and poets to look towards Germanic Scandinavia rather than Greece and Rome for inspiration (von Herder 1796, p. 488)—and the brothers Jacob and Wilhelm Grimm, who relied heavily on Nordic sources for their recovery of a German(ic) *Volksgeist* from the vestiges of rural traditions, mythology, and folktales. This new Romantic paradigm contributed not only to the construction of separate national identities, but also to the creation of a hazy and particularly polemic concept of 'the North', which was supranational and clearly juxtaposed to 'the South'.[6]

These developments led to a more positive appraisal of northern culture in general and initiated on a more local level what can be called a Frisian infatuation with the North. In the first half of the nineteenth century, the educated elite of Friesland developed an intellectual infrastructure of learned societies and periodicals—in part to fill the void after Friesland's only university, the University of Franeker, was closed by Napoleon in 1811—in which

the Frisian language and Frisian antiquity constituted the main topics of discussion. The Romantic type of Frisian nationalism that took shape in these intellectual circles, heavily influenced by German schools of thought (Oppewal et al. 2006, p. 62), has been interpreted as a direct reaction against the emergence of a thoroughly centralized Kingdom of the Netherlands (est. 1815) and the cultural and political marginalization this process entailed for Friesland (Breuker 2014, pp. 28–46; Breuker 2018). Quite naturally, these intellectuals nourished a nostalgic longing for better times, a primordial age of splendor, when Frisia was still free and not yet subdued by invaders from the South (Poortinga 1962, pp. 7–15; Gezelle Meerburg 2001). In the spirit of the Grimm brothers, middle- and high-class city dwellers were convinced that vestiges of this undiluted, primordial Frisian culture could still be found *outside* the city walls, in the countryside, with its ancient rural traditions and its supposedly uninterrupted oral transmissions (Jensma 1998, pp. 21–22, 37–66; Halink 2021, p. 360). In the Romantic, paradoxical anti-urbanism of the urban-based educated elite, the Frisian countryside came to represent the authentic North, a rural idyll, whereas cities and towns—and especially the provincial capital of Leeuwarden—were increasingly interpreted as bastions of *southern* (Dutch) import culture: *Fremdkörper*, in which the Frisian language and true Frisian character had been subdued (Peverelli 2019; van Hout and Peverelli 2019). One might argue that this ideological opposition between town and countryside led to the spatial synchronization of (a glorified and pastoral) past and (an abhorred, urban) present in the Frisian mind, experienced by many as a painful dissonance.[7]

### 3. Frisian or Dutch?

In the Frisian and Dutch national discourses of the nineteenth century, opinions differed on the position of Frisian national culture vis-à-vis the greater construct of Dutch national identity; were these two mutually exclusive national characters, or was Frisian identity a regional manifestation of Dutch national identity? In his contribution to the 2018 *Encyclopedia of Romantic Nationalism in Europe*, Philippus Breuker wonders whether Friesland can even be considered a national community at all. He establishes that Frisian identity transcends the limitations of provincial identity and is ethnolinguistic in nature, but also that the aspirations of the Frisian movement are more regionalist, less 'national' maybe than those of most cultural communities included in the encyclopedia, due to the strong cultural, historical, and political ties with the rest of the country and low support for independence (Breuker 2018, p. 943).[8] The Frisian movement hardly ever showed signs of separatist aspirations, and Frisians have always been keen participants in both the cultural and the political scene of the Netherlands, often effortlessly combining their Dutch national identity with pride in their Frisian roots. Most Frisians were Protestants, just like the Hollanders (but unlike the predominantly Catholic and hence culturally more divergent provinces of Brabant and Limburg in the south). The fact that the Dutch royal family descended directly from the lineage of Frisian Nassau stadtholders further encouraged the integration of Frisian culture and Frisian history into the *grand narrative* of the Dutch nation.

Both in Friesland and in the rest of the Netherlands, there were many who saw Frisian culture as "one variation within the Netherlandic palette, and even as a more pure and unadulterated manifestation of the national character of the Netherlands-as-a-whole than as something opposed to it; thus strengthening rather than contesting Romantic nationalism in the Netherlands" (Breuker 2018, p. 943). As in the case of Britain, where Scottish Highland culture was at times celebrated as the most undiluted form of British national character, the rural and traditional North was imagined as a place of national *authenticity*; an authenticity which had been lost in the more modernized and urbanized regions of the country. The Frisians of yore were increasingly envisioned as the primordial precursors to the Dutch nation, not least because the name of the ancient tribe which traditionally held this honorary position, the Batavians, had become politically contaminated due to its association with the revolutionary Batavian Republic (Halink 2021, p. 357).[9] Together with the Franks and the Saxons, the Frisians came to constitute the "tribal trinity" out of which

the Dutch nation had allegedly evolved (Beyen 2000). Hence, non-Frisian intellectuals increasingly came to interpret Frisian heritage as part of their *own* cultural identity; the Amsterdam-born Frisophile, poet, and playwright Arent van Halmael used themes from both Frisian and Hollandish history in his overtly patriotic plays, and the liberal statesman Johan Rudolph Thorbecke, generally considered the architect of parliamentary democracy in the Netherlands, was moved by Romantic notions of the Germanic North as the cradle of egalitarianism and parliamentary democracy to study Old Frisian law texts (Breuker 2018, p. 954).

However, not all Frisian intellectuals subscribed to the conflation of Frisian and Dutch national character or to the notion of Frisian culture as a mere variation within the 'Netherlandic palette'. Especially for those who fostered anti-Holland sentiments and resented the surrender of Friesland's historical autonomy—which it had enjoyed in the era of the Dutch Republic—to the new centralized Kingdom, the adoption of alternative identity models could generate new narrative molds in which to pour the 'heroic' Frisian past. In nationalist circles, emphasizing the *contrasts* between Dutch and Frisian culture, language, and national character, through the cultivation of historical culture (Leerssen 2006), became a popular form of cultural activism. In this turn away from the South, bonds of kinship with the other Frisian lands beyond the Dutch borders, in the German lands and even in southern Denmark, but also with England—Frisian and Old English were, and are still considered very closely related sister languages—and with Scandinavia, gained in significance. Frisian intellectuals participated in the Europe-wide network of corresponding scholars and intellectuals, and Montanus de Haan Hettema, a leading figure in the Frisian movement of the early nineteenth century, corresponded with the famous Danish linguist Rasmus Kristian Rask, who would publish his grammar of the Frisian language in 1825. Justification for the cultivation of cultural ties with Scandinavia could be found in the prestigious work of Jacob Grimm, who in his authoritative *Deutsche Mythologie* (1835) proclaimed the following:

> Die Friesen bilden in jedem betracht den übergang zu den Scandinaviërn; bei dem vielfachen verkehr dieser beiden an einander grenzender völker ist nichts natürlicher als de annahme, dass den heidnischen Friesen auch die gewohnheit des tempel- und bilderdienstes mit jenen gemein war. (Grimm 1835, p. 79)

> The Frisians constitute in every respect the transition to the Scandinavians; considering the manifold connections between these two neighboring peoples [*völker*], nothing is more natural than the assumption that the pagan Frisians also had the habit of temple and image worship in common with them.

Nine years later, the Dutch librarian and publicist Derk Buddingh quoted this passage in his *Verhandeling over het Westland* (Treatise on the Westland; 1844) and asserted that unlike the Batavians, who were closely associated with the *Germanen* (associated with what is now Germany), the Frisians were more intimately linked with the Saxons and the Scandinavians (Buddingh 1844, p. 176).[10] This essential distinction between Frisians and Batavians, anachronistically applied to the present as a distinction between the province of Friesland and the rest of the Kingdom, could be used as a tool for cultural self-contrastation and for fleshing out a distinctly 'un-Dutch' and exclusively Frisian self-image. Paradoxically, this antagonistic stance entailed resistance against non-Frisian appropriations of Frisian heritage, while at the same time adopting and internalizing the romanticization initiated by those very same outsiders.[11] It has to be emphasized here that this process of cultural juxtaposition rarely led to separatism or calls for Frisian independence. The economic, cultural, and political ties with the kingdom remained strong, just like the awareness that a fully independent Frisian state would be unrealistic and undesirable to most.

On the basis of the linguistic similarities between Old English and Frisian, Frisian intellectuals such as Joast Hiddes Halbertsma, one of the key figures in the Frisian movement of the first half of the nineteenth century, could speak of an 'Anglo-Frisian race', which was considered quintessentially Nordic (and hence very different from German and

Dutch culture) on the basis of his belief that both the Frisian and Anglo-Saxon languages evolved in the area roughly between the river Elbe in northern Germany and Danish Jutland (Halbertsma [1836] 2021). This linguistic discourse has certainly contributed to both the English and the Nordic orientation of the Frisian movement throughout the nineteenth and twentieth centuries.

One of the first Frisian authors to actively engage with Scandinavian culture in his quest to purge the Frisian language of Dutch influences was the school teacher and writer Harmen Sytstra, who in 1845 founded the literary journal *Iduna*.[12] Sytstra was, like many Frisian intellectuals of his generation, fascinated by the language situation in Norway, where the nationalist Ivar Aasen sought to replace the heavily Danified form of Norwegian (*Bokmål*) with his own 'reconstructed', more 'authentic' form of Norwegian called *Landsmål* (van Elswijk 2012). In his journal, Sytstra introduced his readers with zealous enthusiasm to his own archaicized version of Frisian (the so-called *Iduna spelling*), based on Old Frisian vocabulary and spelling and containing many appropriate neologisms to replace words which he deemed too Dutch (van Elswijk 2012; Halink 2021, pp. 361–63). Sytstra's linguistic purism dovetailed with a longing for primordial authenticity, and the very title of his journal may serve as an indication as to where he believed he had found this. The Old Norse goddess Iðunn (Iduna), divine guardian of the apples of eternal youth and hence associated with rejuvenation and poetic beauty, had already become a potent symbol of national regeneration elsewhere and given her name to the journal of the influential Geatish Society in Sweden, among other things. As an emblem of reinvigoration, she could be appropriated by national movements and literary societies throughout Europe. Sytstra became very enthusiastic about Norse mythology after having read a concise introduction to the topic for children (Michiel Jan Noorde-wier's Overzicht der Noordsche Godenleer from 1842) and named his journal *Iduna*—in part also inspired by the short-lived Flemish journal *Wodana* (1843), founded in Ghent by Jacob Grimm's student Johann Wilhelm Wolff—even if this goddess had, in all likelihood, been unknown to his Frisian ancestors:

> …wy mienden, howol wy næt foar der fúst wei bewíse kenne, det Iduna alear by da Friesen forere wirden is, út der lykheid, der wy twiske da fryske end noardske godenlear fine, derta bislúte to meyen, end dos ac frydom to habben, ús wirkje nei hir to neamen […] Friesen! nei socken godinne neamden wy disse bledtsjes, hit stiet oan jimme as wy dermei ta ús doel komme. Hwet det doel is, hoeft nu hast næt ienris mear seid to wirden. (Sytstra 1845, pp. 35–36)

> …we considered, although we cannot easily prove that Iduna was worshipped by the Frisians of yore, that we could, on the basis of the similarity between the Frisian and the Nordic lore of the gods, permit ourselves and thus to have the freedom to name our work after her. […] Frisians! We named these papers after such a goddess, it is up to you if we will thereby reach our goal. What this goal consists of hardly needs any further clarification.

This observation in the first volume of the journal led one of its readers to the hypothesis that traces of the goddess could actually be found in Frisian placenames such as Idaarderadeel (Sytstra 1846, p. 64), which may be considered illustrative of the wish to link Frisian antiquity to the cultural sphere of Scandinavia. Together with Tiede Roelofs Dijkstra, with whom he had founded the Society for Frisian Language and Literature (*Selskip foar Fryske Tael en Skriftekennisse*) in 1844, Sytstra employed a profoundly Nordic frame of reference in his systematic analysis and reconstruction of pre-Christian Frisia (van der Molen 1973). With his journal, Sytstra acquainted his audience with mythological narratives from the Eddas while maintaining that the Frisian gods—such as Wéda, the West-Frisian equivalent of Odin/Wodan—were merely local variations on the same Nordic theme. He also provided his readers with the first ever Frisian translation of a Scandinavian text, being the story of Thor and the giant (or *jötunn*) Útgarða-Loki from the Old Icelandic *Prose Edda* (van Elswijk 2014, p. 14). Sytstra's mental conflation of Frisian and Scandinavian antiquity constitutes the first stage of what I will call the *nordification* of the Frisian self-image, which

served the same purpose as Sytstra's linguistic agenda: to accentuate the contrast between 'Nordic' Friesland and 'southern' Holland.

Although not directly concerned with Old Norse culture as such, the notorious *Oera Linda Book* (which first appeared anonymously in 1872) did much to forge a link between Nordic mythology and Frisian identity as well. This forgery, or hoax, posing as an ancient manuscript written in runic letters, portrays the history of the ancient Frisians from the Creation to approximately 50 BCE, evoking a matriarchal and democratic society that flourished before the Deluge. Before these Frisians laid the foundations of all great Indo-European cultures of the ancient world, their progenitor—a matriarch named Frya— explained to them what it means to be Frisian, or a "child of Frya": "[O]nly he who is not a slave of someone else or of his own passions I can acknowledge as Fryas" (Jensma 2004, pp. 32, 92–93; Oera Linda 11/15–17). This piece of playful etymology not only suggests an essential link between the terms 'Frisian' and 'free' (to which I will return later), but also with the figure of Frya. It is very likely that one of the main suspects of this forgery, the poet and theologian François HaverSchmidt (Jensma 2004), found inspiration in Derk Buddingh's hypothesis that the goddess Freya was considered 'mother of the Frisians' (*mater Frisiae*) until as late as 1303, after which the Virgin Mary inherited the title (Buddingh 1844, p. 71).

## 4. Reimagining Redbad

Around the turn of the twentieth century, this Romantic nordification had had a profound effect on the way Frisian intellectuals conceived of their own national character. A telling example of this can be found in a public lecture by the famous Frisian poet and socialist politician Pieter Jelles Troelstra, which he delivered in January 1910. In his musings on the link between poetry and national character, Troelstra compares the Frisian to a rugged Icelandic landscape, which may at first seem rough and inhospitable, but where, beneath the surface, all sorts of things are moving and glowing. There is more to a Frisian than meets the eye, just like in the case of Iceland's barren volcanic landscape, in which geysers "nu en dan even opspringen uit de grond, om met hun hete dampen te getuigen van het warm leven, dat er onder brandt en werkt" (Troelstra 1910; sporadically erupt through the surface, to testify with their hot steam to the warm life, which burns and stirs underneath). Surely, comparing national character traits to landscape features was nothing new in Troelstra's time and had become something of a commonplace in the national Romanticism of the previous century. However, in virtually all national discourses, it was the landscape in which the nation in question was itself embedded, the features with which it identified, with which it was intimately familiar and inextricably intertwined, and which were believed to have shaped its peculiar characteristics, which served as a metaphor for its *Volksgeist*. Troelstra however does not refer to the endless green flatness of the Frisian countryside, but to the geological splendors of a subarctic island, far removed from the world he and his listeners were familiar with. And still, the metaphor seems to have worked. Given the strong sense of kinship vis-à-vis the Nordic world, popularized and cultivated by Frisian intellectuals such as Harmen Sytstra in the preceding century, the island of saga literature and Viking Age grandeur may have been more familiar to an educated Frisian audience, or less alien at least, than one might expect. I would argue that for this reason, the association with a quintessentially Nordic landscape worked better for the rhetorical and emotive purposes of Troelstra's lecture than the association with the foreign landscapes of, say, a *southern*, Mediterranean character.

This observation brings us to the topic of *climatic determinism*, or the popular idea that collective identities and national characters are organically shaped by the climatic and geographical conditions in which they emerged and evolved. In short, it considers peoples the product of their surroundings. Rooted in the Enlightenment ideas of Montesquieu and Kant, later coupled with popular concepts of social Darwinism and racist stereotypes, this philosophy tends to propagate the superiority of 'northern' national characteristics over 'southern' ones, because it was only in 'the North' that the spirit of the people

had been hardened by the elements and thus elevated to a higher level of nobility. For obvious reasons, these popular conceptions served as the perfect intellectual ammunition for national activists in cultural communities that self-identified—or *branded* themselves— as 'Nordic' or 'Northern' (Beller 2007; Leerssen 2019). In the case of Friesland, we can discern this normative dichotomization for instance in a folktale about King Redbad, as recorded by J.P. Wiersma in his collection of Frisian myths and folktales from 1937. In his rendition of the story, Redbad sent a group of spies—including his own son—to the south, to determine the military positions of their Frankish enemies. When these Frisian men left their fatherland, each one of them was still honorable and "plichtsgetrouw, zoals het een West-Fries betaamt" (dutiful, as befits a West-Frisian). However,

> ...toen ze zuidelijker kwamen en de zon hen verwarmde, vergaten zij hun opdracht. Ze gaven zich over aan de drank die de wijngaarden in het zuiden voortbrachten en de liefde van donkere vrouwen. De zoon van Radbod, die de spionnen aanvoerde, stuurde een bericht naar zijn vader dat ze besloten hadden voorgoed in het zuiden te blijven. (Wiersma 1937, p. 49)

> ...as they moved further towards the south and the sun warmed their skin, they forgot all about their mission. They surrendered to the wine that the vineyards of the south supplied and to the love of dark women. The son of Radbod, leader of the mission, sent a message to his father stating that they had decided to stay in the south forever.

Redbad (or Radbod) was so ashamed of his son's negligence that he declared him and the other spies dead to him, as has remained the custom "tot in onze dagen bij de West-Friezen [...] de zoon die zijn plicht verzaakt als dood wordt beschouwd" (Wiersma 1937, p. 49; up to our own days, that West-Frisians [...] consider the son who forsakes his duty dead). The message is clear; the South is decadent, weak, and hedonistic and has a corrupting effect on the noble character of northerners.

In the historical narrative taking shape in the course of the late nineteenth and early twentieth centuries, Redbad became a central and emblematic 'figure of memory' (Jan Assman's *Erinnerungsfigur*): a nodal point in the community's cultural memory, accumulating ever more positive associations with primordial Frisianness and eventually becoming a symbolic embodiment of the Frisian national character itself.[13] The savage heathen, demonized by the church for centuries, could now be rehabilitated as a *bon sauvage*, an idealized image of mankind in its primordial, authentic state, which—in the influential philosophy of Jean-Jacques Rousseau—could never be considered 'evil' (Halink 2020, p. 107). The *topos* of the noble savage took flight in the secular discourse of Romantic nationalism, creating the intellectual framework for the cultural veneration of prolific pagan characters from the nation's most distant past.[14]

Redbad is an exceptionally good example of this modern paradigm shift; the link that medieval chroniclers had forged between him and 'Denmark'—a container term, referring to the barbarous and heathen North (Nijdam 2009)—in order to demonize him and to create a distance between him and the Frisian ancestors, could now be repurposed and cultivated in a positive sense. The moral compass had turned in favor of the North, and the 'good guys' of the medieval sources (the Franks and their missionaries) could now be portrayed as *southern* intruders, the 'bad guys' with an alien creed, facing staunch opposition from their northern adversaries, fiercely committed to safeguarding their ancestral traditions and their original state of freedom. In this narrative template, the Carolingian Franks of Redbad's time are easily replaced with the Hollanders, or with any other southerners encroaching on that which is considered authentically Frisian.

Nowhere is this highly antagonizing interpretation of Redbad and his 'mission' more tangible than in a treatise written by the legal historian Arian de Goede, published in 1936. De Goede did not hail from the province of Friesland, but he considered his own region (in the province of Noord-Holland) part of greater Frisia and hence part of the realm once ruled by "the greatest king the Frisian people ever had" (de Goede [1936] 2018, p. 11). In

his pompous reinterpretation of Redbad's story, the king's fight against the Carolingians is sublimated into an epic clash between two universal and irreconcilable principles: a cultural struggle of cosmic proportions and significance between the freedom and honor of the Germanic "northern realm" on the one hand and the feudalism and militarism of the intruding "southern realm" on the other (de Goede [1936] 2018, p. 43). De Goede, still a student in his twenties when he wrote this work, portrays Redbad as a freedom fighter of international alure, who through his enforcement of the pagan faith and his courageous stance against the Franks slowed down the advance of southern feudalism and safeguarded the "Frisian-Germanic freedom" for the sake of *all* of mankind (de Goede [1936] 2018). The antagonistic cultural determinism underlying this entire treatise—de Goede had already become a fervent supporter of National Socialism at this point—was to have a long-lasting effect on the way Redbad has been imagined.

Redbad's presumed connection with Denmark was already thematized in the historical tragedy *Radboud de tweede* by Arent van Halmael (1839) and would become an ideologically more significant element in the twentieth- and twenty-first-century reception history. Emphasizing the Nordic qualities of this *pars pro toto* of the Frisian nation automatically entailed the reiteration Friesland's cultural distinction from the Hollandish South. Hence, the nordification of Redbad served a very specific ideological purpose. In 1926, one of the intellectual leaders of the Frisian movement, Geert Aeilco Wumkes, explained why the Frisianized version of the king's name (Redbad) should be favored over the Dutch version (Radboud):

> *Redbad*! Sa hat men [...] de namme to skriuwen. Net *Radboud*. Gjin Hollânsk ef Frankysk for him. Né, foars, koart as twa slaggen mei de hammer op in ambyld, hird as de nammen út ´e Edda en de Yslânsaga´s: Snorri, Ragnar, Grettir, Njal. Syn tiid hat de greate tiid west fen Fryslân. Do wier de spankrêft it heechst, de geastkrêft it sterkst. (*italics original*. Wumkes 1926, p. 32; Knottnerus 2021, p. 60)

> *Redbad*! That is how [...] the name should be written. Not *Radboud*. No Hollandish of Frankish for him. No, strong, short like two strikes of the hammer on an anvil, hard like the names from the Edda and the Icelandic sagas: Snorri, Ragnar, Grettir, Njal. His age was the great age of Friesland, when its resilience and spiritual power were at their highest.

Interestingly, the argument in favor of the Frisian name is purely esthetic and emotive, rather than academic; only its supposed acoustic similarity with epic-sounding names from Old Norse-Icelandic literature suffices here, because it is this Nordic character that sets him and 'his nation' apart from the rest of the Netherlands. However, Redbad's heathen character posed a serious problem for Wumkes, who also served as a minister in the Dutch Reformed Church. He attempted to lift this ideological dissonance by claiming, somewhat paradoxically, that the Frisians "knibbelje for it krús fen Golgotha" (kneel for the Cross of Golgotha) nowadays, but that "for it ûnthâld fen ús oarsprongen bliuwt Redbad by ús in eare as in sinbyld fen ynfrysk wêzen" (Wumkes 1926, p. 42; Knottnerus 2021, p. 60; for the memory of our origins, Redbad remains in high esteem for us, as a noble symbol of primordial Frisianness).

In other adaptations of Redbad's story, his pagan worldviews are considered less problematic. The author and Frisian nationalist Douwe Kalma, founder of the *Jongfryske mienskip* ('Young Frisian Movement', est. 1915) sought to emancipate the Frisian language and to strengthen its status as a language of culture and literature and wrote a series of five historical plays based on the lives of the Frisian kings, in imitation of Shakespeare's tragedies (1920–1951). Kalma portrays Redbad as a powerful protector of the principle of Frisian freedom and the last great champion of Frisian heathendom. What this heathendom may have consisted of exactly remains vague, but some of the deities and pagan motives that are invoked in the play and its opening poem—including Bragi, the Norns, and the Alfather—are of a profoundly Old Norse, Eddic nature (Kalma 1949).

The association with Scandinavian antiquity is even more tangible in the more recent historical novel *Rêdbâd. Kronyk fan in Kening* (Rêdbâd. Chronicle of a King, 2011) by Willem Schoorstra. In this popular work, preceded by a Frisian translation of stanza 76 from the Eddic poem *Hávamál*,[15] Redbad has become Wodan's chosen one, fighting with the passion of a *berserker* against the intruding Franks and their intolerant missionaries. Schoorstra's pagan sympathies are nowhere concealed in this book, and the link with pagan Denmark is made very explicit. In Schoorstra's version of the narrative, Redbad's mother is of Danish origin, and the king gets to learn a great deal about his own closely related heathen traditions while spending time in the company of a related Danish chieftain. The fact that the Danes persevere in their pagan ways and expel any Christian missionaries from their lands—unlike Redbad's own father, Aldgilles, who is not ill-disposed towards the promulgators of this new faith—is especially welcomed by the young man:

> Geen voet krijgen de christenprekers daar aan de grond. Zij die de leer van het kruis willen verspreiden, worden weggejaagd [...]. De Denen zien de prekers als vijanden, en die vallen niet onder het gastrecht. Onze vrienden hebben het bij het rechte eind! Hun hoop en vertrouwen stellen ze op de goden van de voorvaderen. Zoals het hoort. (Schoorstra 2018, p. 207)

> The Christian preachers cannot set one foot on this soil. Those who want to spread the teaching of the cross, are scared off [...]. The Danes consider these preachers their enemies, and those are excluded from the right to be treated as a guest. Our [Danish] friends are right in doing so! Their hope and trust are focused on the gods of their ancestors. As they should be.

This positive reinterpretation of the medieval association with the 'barbaric North', initially employed to turn Redbad into a *non*-Frisian, is now cultivated to emphasize the supposedly Nordic character of Frisian culture. By modelling Redbad and his Frisians after the archetype of the ecstatic Viking warrior, or berserker, the very symbols of northern pride and resilience, Schoorstra endows the Frisian past with an aura of Old Norse heroism (Halink 2020, p. 117).

## 5. Free, Frisian, and Viking!

To fully appreciate the significance of Redbad as a Frisian *Erinnerungsfigur*, it is of the essence to situate him in the larger context of Frisian cultural memory. Inherent to the nature of national discourses is a fixation on past golden ages, on better times, when the land was still free and its primordial culture not yet compromised by foreign intruders and their alien creeds. This fixation naturally dovetails with calls for national renewal, and a desire to restore the historical conditions of those glorified golden ages in the present (Smith 1997). In the case of Friesland, two different historical episodes became the focal point of national veneration, one being the age of the Frisian kings—and of Redbad in particular—and the other being the medieval era of the so-called 'Frisian Freedom' (Fryske Frijheid: c. 1101–1498), when the Frisian lands enjoyed a unique and privileged position withing the Holy Roman Empire, characterized by the absence of feudalism and serfdom, while local elites—answering to the Emperor alone—ruled their loosely associated territories amongst themselves (Vries 2019). The legalistic concept of 'Frisian Freedom', initially referring only to this very particular political situation in the Middle Ages, acquired multiple levels of emotive significance in the centuries following its demise, becoming associated with popular slogans such as "Leaver dea as slaef" and "Frysk en frij" ("Rather dead than slave" and "Frisian and free", respectively). It became internalized as a timeless character trait of all Frisians, and eventually—around the turn of the twentieth century—reached the status of a metaphysical concept, constituting the very heart of the 'Frisian spirit' (Frieswijk 2003; Jensma 2019). The attractive alliteration of 'Frisian' and 'freed(dom)', strengthened by the suggestion of an etymological link between the two terms—as expressed in the *Oera Linda Book*—elevated the love of freedom to the all-pervasive core element of the Frisian national character and the essential red line running through *all* of Frisian history. In 1836,

Joast Hiddes Halbertsma had already glorified the ancient Frisians' love for freedom and fatherland, two names with the same significance (Halbertsma [1836] 2021, p. 191). This metaphysical conception of Frisian freedom strengthened a sense of unbroken historical continuity, exemplified by Douwe Kalma's definition of the Frisian movement of his own time—the 1930s—as "de foartsetting fen de Fryske striid om frijheit en rjucht troch de ieuwen hinne" (Kalma 1932, p. 58; Jensma 2018a, pp. 11–12; the continuation of the Frisian struggle for freedom and justice throughout the centuries).

Under the banner of such a vague denominator, reducing all of history to one epic and spiritual quest for freedom, chronology hardly seems to matter anymore. In popular cultural memory, historical embodiments of this primordialized quest for freedom—such as King Redbad, but also the sixteenth-century nobleman Gemme van Burmania, who supposedly refused to kneel for his new ruler because "Frisians only kneel to God" (Hoekstra 2019)—are all conflated into one single evocative narrative, as historical manifestations of the same timeless and universal principle. This becomes clear from a recent panel research study conducted by the Frisian broadcasting company (Omrop Fryslân) on the importance of the infamous sixteenth-century warlord, freedom fighter, and pirate *Grutte Pier* ('Big Pier', nickname of Pier Gerlofs Donia) to modern Frisians. The questionnaire was filled out by a response group demographically representative of the province. Seven out of every ten participants considered Pier one of the greatest heroes in Frisian history, the bane of many a Saxon and Hollander, and a national icon to be proud of. But more than half of the participants connected his name to the famous Battle of Warns against the invading Count of Holland (1345), some two centuries earlier, and forty percent of the participants even placed him in the army of King Redbad, no fewer than seven centuries earlier (Omrop Fryslân 2020).[16] This blatant lack of knowledge about historical characters and events appears to be at odds with the significance attributed to them. But such are the machinations of cultural memory; just as in the case of the medieval *chansons de geste*, in which Charlemagne anachronistically delivers the Frisians from King Redbad, epic densification does away with the constrains of chronological compartmentalization and allows for iconic characters, often many centuries apart, to join forces in a creative pastiche of collective memories, a blurry image of the past. Although often frowned upon, and far removed from the more 'historically accurate' representations of history, this phenomenon should in my opinion not be denounced in favor of a Rankean "wie es eigentlich gewesen" approach, but rather taken seriously as an object of study in its own right and on its own terms, as a genuine emotive experience of the past and a product of a long mnemonic development.

The extent to which these divergent episodes from Frisian history are conflated can also be illustrated with a scene from a historical novel on the life of Grutte Pier, written by the aforementioned author Willem Schoorstra. In this work, Schoorstra continues the line he set out in his Redbad novel and revisits the themes of foreign oppression and violent resistance. The author's previous—and current—fixation on the Early Middle Ages, the golden age of pagan Frisia, clearly colors his representation of this much later era; in one of the book's more graphic scenes, a monk, simultaniously fascinated and appalled by the merciless warlord, is persuaded to drop his Christian reservations and to carve, with a dagger, runes of war into the flesh on Pier's chest (Schoorstra 2015, pp. 72–74). This scene has no grounding in historical sources and can be considered purely fictional, intended only to appropriate and extend the heroic elan of the 'berserker warrior' to the early sixteenth century and to project it onto the most antagonistic, anti-Hollandish icon of the Frisian repertoire. In a way, Schoorstra thus does to Pier what earlier authors had already done to Redbad: he transforms him, posthumously, into something of a Viking.

Especially in the popular imagination, the metaphysical concept of Frisian freedom is linked to the image of the Viking, or the berserker, fighting off anyone who might jeopardize this hard-won freedom in a trance-like fury. The attestations of this Frisian Viking trope are usually found in the form of personal anecdotes, sometimes presented at the beginning of articles or presentations as a point of departure for critical engagement with, and correction of, common misconceptions regarding the past. In her televised lecture

on Frisia in the Viking Age, Nelleke IJssennagger-van der Pluijm opens with a reference to familiar stories about wanton Frisians who, after a couple of drinks in the pub, tend to brag about their heroic Viking origins. As a specialist in Viking history, and in Viking Age Frisia in particular, IJssennagger-van der Pluijm has lost track of how often she has been approached by Frisians who had had a DNA test done, proving that 'Viking blood' runs through their veins (IJssennagger-van der Pluijm 2021). Casually bringing up the topic of this article in the company of Frisians can already be considered a kind of fieldwork, frequently eliciting insightful anecdotes. Recently, a Frisian contact informed me of how his son, preparing to move to Italy, was advised by his friends not to forget his Viking helmet.

Of course, the common stereotype these anecdotes refer to has little to do with the actual Vikings of history, confined to a very particular era in Europe's distant past. Rather, this stereotype should be seen as a fictional embodiment of praiseworthy character traits which many a modern Frisian readily identifies with: masculinity, sturdiness, pride, resilience, self-reliance, industriousness, rebelliousness, and stubbornness. We will now turn to the various ways in which these vague Nordic sentiments have been mobilized rhetorically, especially in the political arena.

## 6. Alternative Identity Templates

In the first half of the twentieth century, the urge of Frisian intellectuals to identify with Scandinavia intensified and eventually culminated in the work of the aforementioned author Douwe Kalma, leader of the *Jongfryske Mienskip*. In his greater vision of the continent, he saw Friesland as part of a supranational 'North Sea culture' and as a 'bridge' between what he considered the closely related North-Germanic (Scandinavia) and West-Germanic (Britain) cultures of Europe (Kalma 1916; van Elswijk 2014). Kalma proclaimed that 'Holland' was essentially different from Friesland since it was not part of this 'Anglo-Scandinavian' culture and lacked a 'Nordic national spirit'. He was equally dismissive of 'Teutonic' culture (the German-speaking nations), which, in his view, was characterized by a ridiculous literature and the veneration of everything military (Frieswijk 2018, p. 238; Kalma 1916). This anti-German stance—which he probably inherited from prominent nineteenth-century intellectuals such as Joast Hiddes Halbertsma—would however make way for a growing affinity with the popular *völkisch* and racial ideologies of Nazi Germany (Halink 2021, pp. 366–67).

In this new discourse, the old and stereotypical North/South opposition was reactivated in a more radical fashion. The anti-Hollandish sentiments of many Frisian nationalists quite naturally caused the Young Frisians to gravitate towards the West—Kalma was a diligent translator of Shakespeare into Frisian—and the North, on cultural, linguistic, and ethnic grounds, and to downplay the cultural and historical ties with the Netherlands. It may come as no surprise that Kalma and several like-minded Frisian nationalists welcomed the rise of National Socialism in Germany, and the occupation of the Netherlands in 1940, as an opportunity for Friesland to give rise to an alternative template of identity, more desirable than the Dutch one. At least that is how Kalma, who himself joined a group of Frisian fascists in 1940 and participated in several research initiatives of the SS, experienced it.[17]

The Frisian infatuation with the North, which was certainly not confined to those with fascist sympathies, led to the translation of several Old Norse-Icelandic texts into Frisian. In 1939, the journalist and Scandinavophile Jan Tjittes Piebenga, who would during the occupation take part in the resistance, published his translation of the Icelandic saga of Hrafnkell Freysgoði. One year later, the heraldry expert and Nazi collaborator Klaes Sierksma delivered his translation of the Eddic poem *Völuspá*, the 'Prophecy of the Seeress' (in Frisian: *De foarsizzinge fen de Wolwa*). Motivated by a sense of ethnic affinity, an increasing number of Frisians enrolled at Dutch universities to study Scandinavian languages and literature. They linked the ideals of their own Frisian movement to those of the national movements in Norway and the Faroe Islands, where the 'indigenous' languages (Faroese and Nynorsk) were fighting for official recognition next to the 'alien' language of the ruler, in these cases Danish or 'Danified´ Norwegian (van Elswijk 2014, pp. 12–17; Bouma 1981).

The fact that Nynorsk (the state-sanctioned version of Aasen's Landsmål) received this official recognition in 1885 filled Frisian nationalists with hope regarding the future of their own 'language of the heart'.[18]

The Frisian language struggle reached its climax in the years following the Second World War, especially in the context of the traumatic events of *kneppelfreed*, or 'Bat Friday' on the sixteenth of November 1951. On that day, the police—armed with clubs and water hoses—acted against Frisian activists and innocent bystanders alike, who were gathered in front of the courthouse in Leeuwarden on the occasion of a controversial lawsuit concerning the use of Frisian in the court of law. Although no lives were lost during this 'battle', the event was traumatic enough to provoke a more belligerent war-time rhetoric, and accusations of cruel and imperialistic oppression by the Dutch state were raised. Teake Hoekema, an admirer of Scandinavian culture and the first to acquire a degree in Frisian from the University of Groningen, considered this the right time to forge stronger bonds with other northern minorities, especially with the Danish speakers in German Southern Schleswig (Danish: Sydslesvig). Hoekema, who had taught himself Danish, wrote passionately about the "storm" that raged over West (that is: Dutch) Friesland for the monthly newspaper *Det brændende spørgsmål* (The burning question; Hoekema 1952b), dedicated to the situation of the Danish minority in northern Germany. In his plea to what he considers a nation closely connected to his own through ties of kinship and a shared experience of oppression—he even claimed that the 'totalitarian' language politics of modern Germany were no different than those of Hitler—he describes his people as "the southernmost Scandinavians" and their struggle as being inseparable from those of other Scandinavian peoples (Hoekema 1952a). He provides his readers with an overview of the long-standing cultural ties between Friesland and Scandinavia, harking back to the national movement of the nineteenth century. For the sake of brevity, he justifies skipping the even older ties forged during the Viking and even the Ice Age (Hoekema 1952a). In the following decades, Hoekema continued to look towards Scandinavia for political motivation, so much so that he gave a 1974 treatise on the political situation of the Faroe Islands (Hoekema 1974) the telling title 'The Faroe Islands an Example for Friesland?' (*De Føroyar in foarbild foar Fryslân?*).

The Frisian tendency to identify with the political situation in Schleswig-Holstein, the northernmost of the German *Bundesländer* (federated states), reached a climax in 2021, when the regional political party representing the interests of the local minorities (the Süd-schleswigsche Wählerverband, or SSW) achieved the unthinkable and actually managed to secure a seat in the *national* parliament (*Bundestag*) in Berlin during the national elections. This party is committed to a more equal distribution of wealth in Germany and to ending the negligence and marginalization of this region by the dominant "East" (that is, Berlin), thus improving the standard of life there. In their campaign for the national elections, the SSW sought to emphasize this view from the North by applying Viking imagery, most notably in a short campaign video which attracted ample attention. The video is set in what appears to be a Viking-Age village, where a man in a fitting costume tells the villagers, the inhabitants of 'Nordgard' (in German) about a storm in the East, about how the gods have forsaken them, and that the people of the North did not bring home enough "silver" from the East, even though energy costs are higher and hospital beds sparser than anywhere else in Germany. These frustrations eventually climax in the battle cry "...to the East! To reclaim the blessing of the gods!" At this point in the advert, the party's representative Stefan Seidler steps into the scene, ending the double temporal register of the message with his contemporary appearance and light parlance, saying that he endorses the views of the Viking orator, although not quite in such warlike (*kriegerisch*) terms. Nevertheless, the campaign's logo—a Viking ship with the text *Mission Bundestag* on its sail—and the slogan—'Now comes the North!' (*Jetzt kommt der Norden!*)—resonate with the belligerent message of the video, and are clearly designed to invoke the memory of those earlier, not so very harmless incursions from the North in search of silver.[19]

The mobilization of Viking imagery apparently worked. The campaign capitalized on popular 'northern' self-images and a widely felt sense of marginalization and endowed its mission with an air of heroic seal and epic determination. The overall tone of the message may sound daunting—as any speech in German about taking on the East naturally would—and its nativist atmosphere may also rightfully trigger some alarm bells. However, not all use of this imagery is automatically *völkisch* in nature, and in fact, the party's main objectives include improving the rights of regions and minority groups in Germany, with special attention for the Scandinavian model of cultural and linguistic inclusion (van der Laan 2021). The nativist imagery serves the purpose of emphasizing the party's rebellious, anti-establishment, and steadfast character.

It comes as no surprise then that the Frisian National Party in the Netherlands (*Fryske Nasjonale Partij*, FNP), the Dutch sister party of the SSW, followed this campaign and its outcome with great interest (Omrop Fryslân 2021a, 2021b). Not only do these parties have a common 'view from the North', from a region that is perceived as culturally and economically marginalized by the political heartland, they also have a shared agenda when it comes to the emancipation of cultural and linguistic minorities; as ´fellow Frisians´, the FNP is also very sensitive to the use of Viking imagery, which serves as a potent narrative template for the expression of boreal sentiments. Members of the FNP even went on a fieldtrip to German Nordfriesland to learn from their comrades in arms there, with the intention to join the forces of these German and "Frisian Vikings" (van der Laan 2021). Hence, the association with vague concepts such as 'Viking mentality' and 'northern identity' can stimulate cross-border cooperation between regions and 'peripheries', vis-à-vis their countries' respective political heartlands (The Hague and Berlin, in this case). It is especially in this sphere of pan-Frisian cooperation that the supranational concept of Frisian identity is endowed with Nordic character traits. This is evidenced by the design of the unofficial inter-Frisian flag, proposed by the so-called *Groep fan Auwerk*—a small collective dedicated to the cultivation of a separate Frisian identity and nationality in both Germany and the Netherlands—in 2006, containing among other things a Nordic cross similar to the one on the Icelandic and Norwegian flag. In their explanation of the flag's symbolism, the group justifies this emblem on the basis of the kinship between Friesland and other nations around the North and the Baltic Sea, where this design is common. Reference is also made to the long-standing traditions of egalitarianism, parliamentarism, and classless society—as opposed to 'southern' feudalism, forced upon the North by foreign intruders—which Friesland shares with the Scandinavian countries and with Iceland in particular.[20] In other words, it is the Romantic, primordial *myth* of a utopian, free, democratic, and classless North with which these particular Frisian nationalists seek to align themselves. This sense of kinship is thus clearly born from the "fundamental (now largely outdated) assumption that medieval Iceland had been a retreat for a genuine 'Germanism'" (van Nahl 2022, p. 5). It must be noted however that this particular type of ideological nordification, firmly linked to pan-Frisian cooperation, is indeed a fringe phenomenon and not very wide-spread beyond the circles of Frisian national activists such as the Groep fan Auwerk. Nevertheless, the inter-Frisian flag (or the *North Sea* flag, as it is alternately known) is nowadays not an uncommon sight at regional events and commemorations.

Political mobilization of popular Viking sentiments serves the purpose of polarizing the political debate, as a rhetorical device with which to accentuate a cultural—and deeply normative—North/South, center/periphery dichotomy. This is not only the case in the election campaigns of official parties; whenever political decisions taken in The Hague are experienced—by some at least—as being at odds with the interests of Friesland, one will read slogans such as "Don't mess with the Frisian farmers. We have Viking blood" on social media, in memes, and on protest banners (Knottnerus 2021, p. 70). These displays of northern otherness are often accompanied with (images of) the historically incorrect horned helmets: visual markers of a heroic, deviant character and arguably the most 'banal' expressions of boreal identity (Billig 1995). The same archetypal significance can be contributed to the Viking ship. In his preface to the catalogue to the aforementioned

Viking exhibition, the director of the Frisian Museum, Kris Callens, describes how Frisian students had built a replica of a small Viking-Age war ship, which would be assembled and ready for sailing after the exhibition. Callens is excited about the prospect of a Viking ship sailing on Frisian waters again, after all those centuries, and claims that upon this sight "we will all become a bit Viking again" (Callens 2019). It is worth mentioning that this fixation specifically on the figure of the Viking is a very recent, post-2000 development, which seems to coincide with the ´New Viking Wave´ to which I will return in the next chapter. This popular self-image differs significantly from the ways in which the north of the country was portrayed and valued before the turn of the twenty-first century (Jensma 2018b).

Regional identities, especially in peripheral regions, often take shape in a discourse of perceived oppression by the political heartland and give rise to a "partly fictional, pseudo-heroisizing rendition of the past" (van Elswijk 2014, p. 13). The tendency to heroisize the past—for instance by emphasizing a link with Viking warrior culture—is more prominent in smaller nations and cultural communities, due to the defensive position from which their history is related and repurposed, and serves to compensate for their marginalized position in the national discourse (Halink 2017, 2020). According to Goffe Jensma, the tendency to glorify historical manifestations of the community's urge for freedom is a typical feature of identity discourses of minority cultures throughout Europe (Jensma 2019, p. 36).

## 7. The Viking Myth in Popular Culture

Ever since the end of the Second World War, declarations of affinity with abstract identitarian concepts such as 'Nordic spirit' or 'Nordic character' have, for obvious reasons, become highly problematic. An impression of the sensitivities surrounding this topic can be gathered from the reactions to the controversial poem *Nordyske geast* (Nordic spirit; 1978), composed by the poet (and Frisian nationalist) Garmant Nico Visser, which was considered so revolting and inappropriate, coming from someone who had experienced the war firsthand, that literary critics in their indignation wondered whether the poem should actually be read as satire or as an exercise in poetic irony (Sirkwy 2019). However, despite the general dismissal of Nordic sentiments of this now outdated type, a more evocative sense of northern identity and identification with Scandinavia has persevered and has remained a defining element of the popular cultural, artistic, and literary scene in Friesland to this very day. When Jelle Brouwer and his spouse Ina Brouwer-Prakke published their Frisian translations of selected short stories from Scandinavia in 1962, they gave their anthology the title *Frjemd en dochs eigen* (Foreign but still our own): a telling indication of the perceived kinship with the North. Interestingly, the expression of regional cultural identity and the celebration of foreign, Scandinavian art, literature, and culture are still intrinsically intertwined.

The best example of this type of 'new' borealism is the annual cultural crossover festival *Explore the North*, which takes place in Leeuwarden and offers a stage to Frisian and Scandinavian artists, musicians, and poets to 'explore' their affinity with 'the North'. The festival is—in the words of the festival organizers themselves—"firmly rooted in the region, but with openness towards the outside world and searching for connections with the Northern regions" (van Elswijk 2014, p. 11). Placing the aims of this festival in their historical context, Roald van Elswijk notes that, from the Frisian perspective, the need to connect "with other northern cultures is an old one. Initiatives such as 'Explore the North' thus appear to be a relaunch of a historical tradition, a rediscovery and also a reassessment of the cultural ties with Scandinavia" (van Elswijk 2014, p. 11). The modern concept of Friesland as a place of dynamic cultural transmission and a meeting place for different experiences of northernness is to some extent akin to Douwe Kalma's vision of Friesland as a *bridge* between the Scandinavian and the Anglo-Saxon world, albeit completely stripped of its contaminated racial and geopolitical implications.

Also on a less explicit level, the cultivation of boreal sentiments is omnipresent in Frisian culture. To give just two examples: the great ice arena in Heerenveen—ice-skating temple not only of Friesland, but of the Netherlands and arguably even the whole world—

founded in 1967, is called Thialf, a reference to Þórr's athletic servant Þjálfi known from Old Norse-Icelandic mythology. The province's most prolific folk metal band is called *Baldrs Draumar* ('Baldr's dreams', which is also the name of one of the Eddic poems), and it glorifies both the pagan gods of the Norse pantheon and the heroic past of Friesland in its Frisian lyrics.[21]

The latest chapter in the turbulent reception history of King Redbad, and the final stage of his cultural nordification, began in 2018 with the release of the historical action movie *Redbad* (Farmhouse Film & TV), directed and produced by Roel Reiné and Klaas de Jong. This Dutch production, which met with heavy criticism from historians and would become the most expensive box-office flop in Dutch cinematic history, presents the Frisian king in a rather different light than Schoorstra did in his novel. Here, he is no longer a staunch defender of indigenous heathenism, but rather a down-to-earth rational Dutchman *avant la lettre*, a man of the Enlightenment projected back onto Frisian antiquity, who rejects both the religious fanaticism of the Frankish missionaries—a clear reference to contemporary Islamic fundamentalism—as well as that of his own human-sacrificing tribe (Jensma 2019, p. 53; Halink 2020, p. 117). This ideologically charged anachronism is directly derived from Arian de Goede's controversial essay on the king (Goosmann 2018). Like Schoorstra's novel, part of the movie is set in Denmark, where the king reconnects with his own roots. In these scenes, the stylistic resemblance to internationally acclaimed TV series, such as *Vikings* (History Channel 2013–2020), *The Last Kingdom* (BBC/Netflix since 2015), and even *Game of Thrones* (HBO 2011–2019) is striking. Everything down to the haircuts and tattoos, the battle scenes, the filming techniques, and the almost obligatory trope of the 'warrior princess' make *Redbad* very much part of what could be called the 'New Viking Wave' in popular culture, albeit a significantly less successful one.[22]

At this point, the modern reception history of Redbad coincides seamlessly with that of the Viking, following the same track from noble (or fierce) savage with Wagnerian headdress to emancipated 'hipster', embodying our own contemporary values and sensitivities (Birkett 2020). In this remediation of the Frisian past, the conflation of ancient Scandinavia—or at least a popular rendition thereof—and Redbad's Frisia reaches a climax, especially if one considers the movie's (limited) foreign merchandise and visual adaptations beyond the cinema. The Australian release of the movie was—undoubtedly for marketing purposes—renamed *Viking Uprising: Legend of Redbad* and the French one (somewhat confusingly) *Vikings. L´invasion des Francs*. The film poster for this last release even places Redbad in front of an imposing landscape of tall mountains and majestic fjords; very raw and Viking-esque no doubt, but not exactly representative of the landscape in either the Netherlands or Denmark.[23] Whereas the movie itself remained virtually unnoticed in the rest of the Netherlands, some political activists in Friesland did pick up on its Redbad/Viking connection. The text "Don't mess with the Frisian farmers. We have Viking blood", placed over a screenshot from one of the movie's battle scenes, has been spotted as a banner at protests against agricultural regulations from The Hague (Knottnerus 2021, p. 70).[24] Redbad's posthumous career as a Viking is still very much in full swing.

## 8. Conclusions and Reflections

After having charted the origins and evolution of the Frisian Viking sentiment in popular culture and politics and in ideological treatises and social media, it is time to return to the questions posed in the opening pages of this essay. What should we, as scholars, do with all this? Is it our task to counterbalance these popular, ideologically charged and distorted images of the past with solid scholarship? Is it even necessary for historians to take a stance vis-à-vis the modern development of the Viking 'from barbarian to brand' (Dale 2020)? As indicated by Jan Alexander van Nahl in his introduction to the current special issue, there is a "long-standing pessimism of scholars in historical subjects", linked to the "sentiment of losing track of the larger picture, and of having lost significance for the here and now" (van Nahl 2022, pp. 3–4). How can the so-called ivory tower of academia still be of significance in all this? How to bridge the gap between genuine historical and

editorial work on the one hand, and the popular images and 'extravagances' charted in this paper on the other?

The natural first reaction of trained historians when confronted with blatant historical inaccuracies in movies, political rhetoric or in popular culture is the "Rankean reflex" (Jensma 2019, p. 53), which amounts to an evaluation of the cultural product in question purely on the basis of its level of correctness (or lack thereof). Does it do justice to historical reality, to "wie es eigentlich gewesen"? The academic reception of the aforementioned movie *Redbad* was overwhelmingly characterized by this strategy, resulting in exhaustive laundry lists of inaccuracies and the pedantic waving of fingers. When it comes to the assessment of publications or productions with an educational purpose, the Rankean reflex is a useful, indeed a necessary one. It takes a critical eye to notice the more subtle ways in which the Nordic frame of reference can influence modern conceptions of the past in this genre. For instance, in the catalogue to the permanent exhibition of the Frisian Museum, the chapter on Redbad's aborted baptism relates how the ruler suddenly decided that he preferred the prospect of spending his afterlife in *Walhalla*: a term exclusive to the Scandinavian sources and of which no attestations have yet been found in relation to the supposed worldviews of pre-Christian Frisians (van Zijverden [2013] 2014, p. 176). This is worth pointing out because it enhances our awareness of the distortive effects entirely unconscious frames of reference can have on our renditions of the past. However, when it comes to popular culture and products of a creative and artistic nature, when a certain level of poetic license is called for, is there a purpose in pointing out these fallacies? Does it really matter that collective identities are (more often than not) rooted in misrepresentations, or at least oversimplifications of the past?

This is a difficult question to answer, which touches on the moral obligations of the scholarly community as a whole. The northern sentiments analyzed in this contribution have a distinctive *couleur locale* and are determined by a convergence of the historical pull towards the North—away from the Dutch heartland—and an amorphous, emotive concept of primordial, long-lost 'Frisian Freedom'. This may not be a bad thing in and of itself, but historians do tend to foster a healthy suspicion towards nativist phantasies and primordial symbolism, especially in relation to (regional) politics. As we have seen in the case of the German SSW, the political mobilization of boreal themes and Nordic heritage does not necessarily signal a *völkisch* agenda; they may just be employed as 'harmless'—and often ironic—markers of regional/northern/peripheral identity, usually silhouetted against a southern 'other'. This process of self-branding by historical means is as old as humanity itself and should not be judged too harshly in my opinion; it is inextricably linked to the rhetorical nature, the 'game' of center-versus-periphery dynamics.

Things are however quite different when this Nordic repertoire is mobilized in more sinister ways, and the balance shifts from irony to racial essentialism (Leerssen 2019). This was the case in 2019, when the Dutch right-wing populist Thierry Baudet, leader of the party *Forum voor Democratie*, proclaimed that the Netherlands are part of the "boreale wereld" (boreal world), now at risk of being destroyed by those who detest its civilization. In this case, the term 'boreal' is obviously more than merely a geographical indicator, and scholars were quick to point out that Baudet's pompous choice of words was inspired by a long tradition of *völkisch* and racist thinkers and that the term 'boreal' is in this case a euphemism for 'Aryan'.[25] It requires a profound historical awareness and thorough understanding of the history of borealism in modern European culture to be able to distinguish between 'banal' expressions of northern identity on the one hand, and cleverly concealed proclamations of ethnic superiority on the other. This, then, should clearly be considered one of the key tasks of the scholarly community: to apply our knowledge of the past to the political discourses of the present, and to monitor and challenge the excesses of this type of rhetoric, where cultural Nordicism appeals "to those seeking a mythologised home of racial beauty and predominance" (Forssling 2020, p. 223; van Nahl 2022, p. 6).

The cultural proliferation of the Viking stereotype is immensely complex and multi-facetted, and as I have attempted to demonstrate in this contribution, its ideological

mobilization can serve a great variety of different purposes. The image of the Viking can be employed to demarcate *national* identities and to encourage *supranational* cooperation, to connect and to isolate, to act regionally while cultivating a global outlook, to stimulate exclusion or promote inclusion. Rather than immediately opting for a normative approach, I believe it is of the essence that scholars acknowledge this ideological versatility and appreciate the fascinating paradoxicality of 'the Viking' as a mnemonic device. The Viking myth has provided Frisian nationalists with a heroic narrative template, an attractive mold in which to re-cast the Frisian past and, consequently, their self-image. But, more so than many other identity models, the Viking narrative is susceptible to being hijacked by ethnic essentialism and nativist phantasies. Alternative historical templates of 'northern greatness', such as, for instance, the success story of the medieval Hanseatic League, are less problematic in this sense; the concept of 'Hanseatic blood' is as ridiculous as it sounds, whereas the wide-spread misconception that 'Viking' constitutes something of a racial category *does* seem to justify references to 'Viking blood' for rhetorical purposes, for instance by Frisian farmers. It is this essentialist, ethnic aspect of the Viking discourse in popular culture that should be debunked by all means and through all outlets that the academic community has at its disposal.

The 2019 exhibition in the Frisian Museum, *Wij Vikingen* ('We Vikings'), serves as a perfect showcase of how academia and popular conceptions of the past interact in the public sphere. While based on solid scholarship, revealing much about the murky and multi-facetted relations between Frisians and Vikings in a liminal region, the communication towards the intended audiences of the exhibition still actively engaged with the Viking myth in Frisian culture, if only for marketing purposes. In that sense, the exhibition is just as much a reflection of our own time, and hence a worthy topic for historical investigation. Rather than criticizing the distortions in this representation of the past, I would follow Goffe Jensma's suggestion to look at these phenomena—including the universally dismissed *Redbad* movie—as subjects of *positive* scholarly attention (Jensma 2019, p. 53). Rather than pointing out the countless historical inaccuracies, we should ask ourselves *why* the makers decided to depict Redbad in this way, or what motivated the marketing department of the Frisian Museum to promote the Vikings as practically 'family'. When it comes to Viking and Old Norse studies, this contemporary dimension, the study of this material's modern reception history (entailing "various types of appropriation of the past, some of which [...] can be quite ambivalent or even outright opposing when compared"; van Nahl 2022, p. 11), deserves a more prominent position in the curriculum, not least because research agendas—and hence the distribution of research funding—are largely determined by modern sentiments, sensibilities, and (mis)conceptions, which are themselves the result of particular cultural, political, and ideological developments. Creatively recycling the past is an integral part of the past itself. So, rather than pedantically repeating the mantra "Vikings did not wear horned helmets", it would be more useful to consider *why* so many people seem to think that they *did* wear horned helmets, and to chart the *Werdegang* and the cultural significance of this powerful, emblematic idea. We should study the origins and evolution of these popular myths and study the dynamic interaction between the stereotypes and images from *within* and from *outside* the cultural community under scrutiny. This approach would elevate us from the level of simply deconstructing myths, to actually fathoming them and grasping their purpose in society, in the realization that ideas—whether distortive or not—possess agency, and that merely deconstructing myths only creates vacuums for new, equally distortive myths to evolve in (Nijdam 2019). Collective identities do not simply evaporate into thin air once their premises have been proven historically inaccurate. Nor should that be our aim. The more fruitful approach to the study of this reception history entails, in my opinion, a certain respect for mankind's perpetual need for myths, for narrative templates to live by. It should be our task to counterbalance the excesses of this process of mythopoesis with historical nuance and analysis.

One final consideration I would like to share here is that the *regional* dimension deserves more attention in this field of research. Just like in the fields of Viking and Old Norse

studies proper, the merits of a 'glocal' approach (meaning that a deeper understanding of small-scale local developments, and of the global context in which they are embedded, causes their mutual enhancement) are now gaining recognition;[26] the study of the modern reception history of this heritage could potentially evolve into exciting new directions if it could detach itself from its traditional *national* frames of reference and instead evolve an approach in which peculiar regional appropriations and mobilizations of the past are interpreted from a supranational and comparative perspective.

**Funding:** This research received no external funding.

**Institutional Review Board Statement:** Not applicable.

**Informed Consent Statement:** Not applicable.

**Acknowledgments:** I would like to thank Nelleke IJssennagger-van der Pluijm, Han Nijdam, Liuwe Westra, Redbad Veenbaas, Diana Spiekhout, and Willem Schoorstra for their valuable input and feedback on earlier drafts of this article. I am also grateful to Jan Alexander van Nahl and the anonymous reviewers for their useful comments and suggestions. All remaining errors are of course entirely my own.

**Conflicts of Interest:** The author declares no conflict of interest.

## Notes

[1] It is estimated that the exhibition attracted some 88,000 visitors, two-thirds of whom came from outside the province of Friesland (Brugman 2020).

[2] All translations are my own, unless otherwise indicated.

[3] Note on terminology: although the terms Friesland and Frisia are sometimes used interchangeably in reference to the medieval territory inhabited by Frisians—including for instance Ostfriesland and Nordfriesland in Germany—the focus of the present article will be on the modern Dutch province of Friesland/Fryslân.

[4] However, the exact extent of the Frisian realm is still a matter of scholarly dispute. See de Langen and Mol (2020).

[5] Famous though this anecdote may be, it is unlikely that it is based on actual events; see Meens (2014). For a thorough examination of the historical facts and fictions surrounding the enigmatic figure of Redbad, see especially Meeder and Goosmann (2018).

[6] The conceptual history of the interrelated terms 'north(ern)', Nordic, 'boreal(ism)' and 'northernness' is both too long and too complex to disentangle here. All of these terms can carry various layers of significance, depending on the context in which they are employed and by whom. For critical discussions on the ideological/imagological construction of 'the North', see especially Leerssen (2019), Fülberth and Meier (2007), Arndt (2004), Jakobsson (2009), and Fjågesund (2014).

[7] That this dissonance between city and the countryside is still an issue in contemporary Friesland has been demonstrated by van Hout and Peverelli (2019). Along very similar lines, Romantic nationalists in Iceland portrayed Reykjavík as a *Danish* town and a blemish on the authentic 'national landscape' (Halink 2014).

[8] However, given the great ambiguity of the terms national identity and nationalism themselves, I believe that the calls for linguistic emancipation and greater regional autonomy that characterize the Frisian movement certainly justify its classification as a (moderate) national movement.

[9] The Dutch state was known as the Batavian Republic—or the Batavian Commonwealth—between 1795 and 1806, first as a client state of revolutionary France and later as part of the Napoleonic Empire.

[10] For his Frisian sources, Buddingh relied largely on the expertise of Montanus de Haan Hettema—among Jacob Grimm's main contacts in the Netherlands and the first to write a review of his *Deutsche Mythologie*—and Joast Hiddes Halbertsma, one of the leading intellectuals of the Frisian movement.

[11] A recent example of Dutch/Hollandish appropriation of Frisian heritage concerns the so-called *Elfstedentocht* (or 'Eleven cities tour'), the famous long-distance skating event on natural ice along the eleven cities of Friesland, which is described in a video on the website of the Dutch national broadcasting company (NOS) as a "oer-Hollands" (typically Hollandish) phenomenon: https://nos.nl/video/2411244-oer-hollands-en-een-mega-evenement-giet-it-ooit-nog-oan (last accessed: 9 March 2022).

[12] In 1851, *Iduna* became the official journal of the Society for Frisian Language and Literature (*Selskip foar Fryske Tael en Skriftekennisse*).

[13] For the turbulent reception history of Redbad, see especially Meeder and Goosmann (2018), Goosmann (2018), Nijdam and Knottnerus (2019), Halink (2020), and Knottnerus (2021).

[14] This led to the rather paradoxical situation of modern nationalists, usually Christians at least in name, idealizing their pagan ancestors and their struggle against 'foreign' missionaries for ideological purposes. See on this paradox especially Halink (2017, 2020, 2021).

[15]    Some years later, in 2020, Schoorstra published the first integral Frisian translation of the *Poetic Edda*.

[16]    Another forty percent could not say for sure whether or not Pier had fought side by side with Redbad. For further analysis of the reception history of Grutte Pier in Friesland, see especially Raat (2019).

[17]    On the role of the SS and its research institute *Das Ahnenerbe* in Frieland and on the Frisian movement during and between the two World Wars, see Frieswijk (2021) and Zondergeld (1978), respectively.

[18]    This fixation on the linguistic situation in Norway may account for the disproportionally large number of Frisian translations of Norwegian literature in the twentieth and twenty-first centuries, especially in comparison to translations from Swedish, which is a much larger language. See van Elswijk (2014), p. 15.

[19]    This 2021 campaign video is accessible here: https://www.omropfryslan.nl/nijs/1095292-fnp-folget-mei-spanning-dutske-ferkiezingen-sit-foar-susterpartij (last accessed: 7 March 2022).

[20]    Until recently, these arguments could be found on the website of the Groep fan Auwerk (http://www.groepfanauwerk.com (accessed on 8 March 2022)). However, during my work on this article, the website appears to have been taken offline. For more on the inter-Frisian flag and its significance, see the Frisian *Wikipedia* site on this topic: https://fy.wikipedia.org/wiki/Ynterfryske_Flagge (last accessed: 8 March 2022).

[21]    In 2015, the band released an album called *Aldgillessoan* (Aldgilles's son), entirely based on Willem Schoorstra's rendition of the Redbad narrative.

[22]    I use the term 'New Viking Wave' here in a very different sense than Katia Kjartansdóttir (2011), who links it to the use of Viking heritage in the context of expanding tourism and neoliberal ideology in modern Iceland.

[23]    For the film poster in question, see https://video-a-la-demande.orange.fr/film/VIKINGSLINVW0144672/viking-l-invasion-des-francs (last accessed: 10 March 2022).

[24]    Patriotic Frisians are also surfing the 'New Viking Wave' by producing memes with screenshots from internationally acclaimed Viking-themed series. This example for instance, from a Frisian Facebook site against political mismanagement, features an image from *Vikings* in which the shields have been decorated with the Frisian flag: https://www.facebook.com/FRLFRLGH/posts/2357052471234530/ (last accessed: 10 March 2022).

[25]    For a thorough analysis of the speech in question on the website of the Dutch national broadcasting company (NOS), among others by the literary scholar Joep Leerssen, see: https://nos.nl/artikel/2277077-de-uil-van-minerva-en-oikofobie-wat-zei-baudet-nou-eigenlijk (last accessed: 12 March 2022).

[26]    A good example of this approach is the international research project on assembly or *thing* sites throughout the Nordic and British world. See https://www.thingsites.com/ (last accessed: 12 March 2022).

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
