# Peer review of "“Almost Like Family. Or Were They?” Vikings, Frisian Identity, and the Nordification of the Past"

_humanities, doi:10.3390/h11050125_

Round 1
Reviewer 1 Report
The author (A.) certainly has a point by taking the enormous fascination of our own time with the Vikings (a vast literature, but also popular interest in HBO and Netflix shows etc.) as the starting point for an investigation into the representation of the North in the Frisian culture of the 19th until the 21st centuries. This is an interesting and relevant topic from an international as well as from a Frisian/Dutch perspective.
However, the elaboration of this topic is insufficient.
· A. does not does not differentiate sufficiently according to time periods. Sources from different periods are associatively strung together without a robust argumentation to stick them together. Furthermore, I wonder whether the fascination for the North (which certainly is a characteristic of modern Frisian (but also Dutch) culture) can be described so absolutely as 'the nordification of Friesland'. Firstly, Frisian scholars and writers were also part of Dutch cultur, secondly, they focused on a broader European context than just the nordic one and – as A. rightly points out – the Pan-Frisian connections. A portrays e.g. Joost Halbertsma as a champion of Nordic culture whereas in reality the Anglo-Frisian connection is à much more salient feature of his work.
· The author lumps abstract concept such as 'the nordic' or even more abstractly 'the north' together with more concrete concepts such as Scandinavia, Norwegian and/or the Danish. More explanation is therefore needed (also on missing concepts like f.i. ‘North Sea culture’ or Het Noorden van Nederland, ‘The North’ (as a part of the Netherlands).
· The author presents and reasons on the basis of disparate material (newspaper articles, anecdotes, historical sources and scientific literature from the past and the present) without sufficiently indicating the quality of these materials.
All in all, in my opinion may become an innovative contribution, but at the moment is no more than an interesting, but also associatively written article that draws too much of its persuasiveness from rhetorical exaggeration.
Also due to the relevance of the topic and the valuable collection of materials, I would certainly want to encourage A. to revise the article.
As to the used literature, I have a few suggestions as to omissions I signaled. A. missed:
o Seminal archaeological studies exactly on North Sea culture, notably: Nicolay, Johan (Johan Auke Willem), The splendor of power: early medieval kingship and the use of gold and silver in the southern North Sea area (5th to 7th century AD). studies, [ISSN 1572-1760] volume 28 (Eelde: Barkhuis Publishing; 2014) and Bazelmans, Jos, By weapons made worthy: lords, retainers, and their relationship in Beowulf (Amsterdam: Amsterdam University Press 1999).) and in its footsteps he should look into modern studies on the Beowulf as the embodiment of North Sea culture. The important work by Noomen (and in his footsteps Mol and De Langen) on Early Medieval Friesland is also missed.
o A. seems tp have missed Sverrir Jakobsson, Images of the North histories, identities, ideas (Amsterdam; New York, NY: Rodopi 2009)
o A. did not notice the early nineteenth-century interest in Nordic languages by Rasmus Rask and his Frisian correspondent Montanus de Haan Hettema
o A. does not sufficiently show how Frisian culture is part of Dutch culture. Literature he lacks is for example: Beyen, Marnix, 'A Tribal Trinity: The Rise and Fall of the Franks, the Frisians and the Saxons in the Historical Consciousness of the Netherlands since 1850', European History Quarterly 30 (2000) 493–532 .
o The way in which the Fries Museum deals with concepts such as 'Kings of the North Sea' and 'We Vikings' differs from the ways in which the North was portrayed and valued before about 2000; see, for example, Jensma, Goffe., 'Remystifying Frisia. The experience economy along the Wadden Sea Coast', in: Linda Egberts and Meindert Schroor ed., Waddenland Outstanding. History, Landscape and Cultural Heritage of the Wadden Sea Region. 151–167 (S.l.: Amsterdam University Press 2018).
I would want to advise the author to take the current, 21st-century interest and the way in which the contemporary experience industry (such as the Fries Museum) uses this as a marketing tool, as a starting point, from which he then could derive his main question ('what is in Friesland the prehistory of this fascination with North Sea culture, Vikings and the North? I would also advise the author to specify his concepts more precisely and to reflect more on the historical development of those particular concepts.
Author Response
Dear Reviewer 1,
Thank you very much for your thorough evaluation of the first draft of this article. This is a typical example of peer reviewing, with one reviewer suggesting major revision, the other one seeing no need for anything but a few cosmetic corrections. This contribution is one of the first in the field to provide an overview of the situation in the Netherlands, and will serve as a promising starting point for further examination of the topic. This is what reviewer 1 seems to suggest, but I do not see the necessity to have all this work done in this single paper. Reviewer 2 rightly emphasizes that the paper's conclusions are measured and appropriately nuanced.
I have followed the editor´s suggestion to focus on the literature suggested by reviewer 1 and to include the relevant titles to the list of references. I have also added in a footnote a more elaborate explanation of the problematic and hazy term "the North". I don't think that it is possible to pin all these concepts down, and, again, it is certainly not the task of this paper to follow up the long history of concepts of "the North". But I do reflect upon this challenge in footnote vi, with regards to individual terms. All in all, the editor of this volume did not see the need for very extensive revisions of the text.
With kind regards,
The author
Reviewer 2 Report
This article aims to outline the cultural and intellectual-historical development of perceptions of Vikings and medieval Scandinavian mentalities in Friesland and the Netherlands, with the goal of understanding how the Frisian identity has been affected by the (especially) nineteenth- and twentieth-century Romanticization of the Viking image. The author then uses this historical and cultural analysis as a backdrop upon which to comment on the role of scholars to counterbalance or, when necessary, rebuke misappropriations of the past in the modern socio-political situation.
The author’s approach to the material fits nicely into a growing scholarly and public-intellectual conversation about modern appropriations of the “Viking” image and cultural identity, while also offering to the conversation the unique perspective of the Dutch/Frisian situation, which is not often enough considered by scholars. The article is diligently researched and well written, and its conclusions and reflections are measured and appropriately nuanced. Note the following suggestions:
1. P. 2, top of page: The sentence beginning “A myth which . . .” does not make a complete sentence in English. This can be easily remedied either by joining the phrase to the previous sentence or by slightly revising the beginning of the sentence.
2. P. 2, middle paragraph: “om the other,” should read “on the other”
3. P. 3 top of page: The sentence beginning “ In this historical narrative. . .” has one too many nonrestrictive clauses for clear reading. I recommend putting the “who in reality . . .” phrase in parenthesis rather than m-dashes.
4. P. 3, bottom of page: The sentence beginning “This new Romantic paradigm did not only contribute to . . .” would read better in English as “This new Romantic paradigm contributed not only to . . .
5. There appears to be some inconsistency in how the formatting addresses block quotes. Most are double spaced while the middle of pg. 3 is single spaced.
6. Bottom of pg. 11: The sentence beginning “Of course, the common stereotype . . .” is an important one but is a bit muddled. Consider revising the problemed phrases to read something like: “ . . . confined to a very particular era in Europe’s distant past, but this stereotype should be seen as a fictional embodiment of praiseworthy character traits which many a modern Frisian readily identifies with masculinity: sturdiness, pride, resilience . . .”
Author Response
Dear Reviewer 2,
Thank you very much for your thorough and positive assessment of the first draft of this article. The 6 suggestions were all very helpful, and have all been incorporated in the second draft of the article.
With kind regards,
The author
Round 2
Reviewer 1 Report
My comments are sent to the editor